# Short-Chain Fatty Acids Modulate Sperm Migration through Olfactory Receptor 51E2 Activity

**DOI:** 10.3390/ijms232112726

**Published:** 2022-10-22

**Authors:** Emanuela Teveroni, Fiorella Di Nicuolo, Edoardo Vergani, Carmine Bruno, Giuseppe Maulucci, Giada Bianchetti, Anna Laura Astorri, Giuseppe Grande, Jacopo Gervasoni, Lavinia Santucci, Marco De Spirito, Andrea Urbani, Alfredo Pontecorvi, Francesca Mancini, Domenico Milardi

**Affiliations:** 1International Scientific Institute “Paul VI”, ISI, Fondazione Policlinico Universitario A. Gemelli IRCCS, 00168 Rome, Italy; 2Division of Endocrinology, Fondazione Policlinico Universitario A. Gemelli IRCCS, 00168 Rome, Italy; 3Department of Neuroscience, Section of Biophysics, Università Cattolica del Sacro Cuore, 00168 Roma, Italy; 4Fondazione Policlinico Universitario A. Gemelli IRCCS, 00168 Rome, Italy; 5Unit of Andrology and Reproductive Medicine, University Hospital Padua, 35121 Padua, Italy; 6Department of Laboratory and Infectious Diseases, Fondazione Policlinico Universitario A. Gemelli IRCCS, 00168 Rome, Italy; 7Metabolomics Research Core Facility, Gemelli Science and Technology Park (GSteP), Fondazione Policlinico Universitario A. Gemelli IRCCS, 00168 Rome, Italy

**Keywords:** olfactory receptors, OR51E2, short-chain fatty acids (SCFAs), sperm migration

## Abstract

The non-orthotopic expression of olfactory receptors (ORs) includes the male reproductive system, and in particular spermatozoa; their active ligands could be essential to sperm chemotaxis and chemical sperm–oocyte communication. OR51E2 expression has been previously reported on sperm cells’ surface. It has been demonstrated in different cellular models that olfactory receptor 51E2 (OR51E2) binds volatile short-chain fatty acids (SCFAs) as specific ligands. In the present research, we make use of Western blot, confocal microscopy colocalization analysis, and the calcium-release assay to demonstrate the activation of sperm cells through OR51E2 upon SCFAs stimulus. Moreover, we perform a novel modified swim-up assay to study the involvement of OR51E2/SCFAs in sperm migration. Taking advantage of computer-assisted sperm analysis (CASA system), we determine the kinematics parameters of sperm cells migrating towards SCFAs-enriched medium, revealing that these ligands are able to promote a more linear sperm-cell orientation. Finally, we obtain SCFAs by mass spectrometry in cervico-vaginal mucus and show for the first time that a direct incubation between cervical mucus and sperm cells could promote their activation. This study can shed light on the possible function of chemosensory receptors in successful reproduction activity, laying the foundation for the development of new strategies for the treatment of infertile individuals.

## 1. Introduction

Human reproduction is a complex sequential process that includes the long-lasting journey of motile sperm through diverse compartments of the female genital tract and the final fusion with mature oocytes in the ampullary part of the oviduct. In recent years, the physiology of heterotopic olfactory receptors has become increasingly important in human reproduction, in particular, related to sperm chemotaxis [1]. Olfactory receptors (ORs) are G-protein-coupled receptors, capable of detecting volatile chemicals that act as odorants [1]. The activation of ORs initiates an intracellular signal transduction pathway triggered by the stimulation of adenylate cyclase III, which leads to Ca^2+^ influx and membrane depolarization thereafter [2].

Firstly, detected in the olfactory epithelium, ORs are subsequently identified in several tissues that include, but are not limited to, the brain, bladder, prostate, and testis [1,3,4,5,6,7,8,9]. However, the physiological consequences of the activation of ORs outside of the nose is still uncertain due to the scarce knowledge of the odorant ligands involved.

As recent studies have shown, the testis possess the richest OR expression outside of the nose [6,9,10]. Late spermatids and spermatozoa have been shown to have a selectively distinct anatomical compartment expression of ORs [10,11,12] suggesting their possible involvement in processes, such as maturation, migration, and fertilization [13].

Spermatozoa chemotaxis is mostly dependent on intracellular calcium release. ORs seem to be among the receptors involved in activating this signaling pathway [10,11,12,14,15]. However, parallel Ca^2+^-independent signaling has been described, as evidenced by a signal transduction cascade independently activated by adenylate cyclase’s second messenger cAMP [12,16]. Of note, the CatSper calcium channel in human spermatozoa seems to be stimulated by odorant ligands too [16]. Neuhaus and colleagues reported ERK phosphorylation as a further molecular signature related to OR-mediated sperm activation. Indeed, they treated sperm cells with the odorant bourgeonal, and they showed an accumulation of pERK upon the odorant stimulus, concomitant with the activation and acrosomal relocalization of the cognate receptor OR17-4 [11]. As a general sign of sperm-cell activation, it is tempting to speculate that this molecular signature underlying the OR activity is related to sperm chemotaxis.

In fact, growing evidence of the close relationship between ORs and spermatozoa chemotaxis is spreading in the current literature. OR17-4 (also known as OR1D2) is localized in the spermatozoon midpiece [11]. In 2003, Spehr and colleagues discovered the aforementioned bourgeonal, a ligand for human OR17-4, capable of increasing Ca^2+^ as well as inducing chemotaxis in human spermatozoa [14]. Although bourgeonal is most likely not an endogenous compound in the human body, the study supported the idea that human ORs act as chemoreceptors for small molecules in sperm. Interestingly, a reduced olfactory perception of burgeonal has been correlated with idiopathic infertility [17].

Other odorants, such as Myrac and PI-23472, ligands of OR7A5 and OR4D1, respectively, have been demonstrated to influence the activation of spermatozoa motility. These same receptors seem to be activated by other ligands found in vaginal secretions and follicular fluid, in particular, 5α-androst-16-en-3-one for OR4D1 and 4-hydroxy-2,5-dimethyl-3[2H]-furanone for OR7A5 [18].

Among the evaluated odorants, great importance has recently been attributed to volatile short-chain fatty acids (SCFAs), the metabolic by-products of anaerobic bacterial fermentation. OR51E2 and its murine ortholog (Olfr78) have been reported to respond to lactate [19] and SCFAs, such as propionate (hereafter PA) and acetate (hereafter AA), in the kidneys [20]. However, OR51E2 is not exclusively limited to the kidneys. Indeed, Aisenberg and collaborators reported OR51E2 expression in airway smooth muscle cells, in which they found a cytoskeleton remodeling mediated by OR-SCFAs binding [21]. Moreover, Flegel and colleagues showed that OR51E2 is expressed on the extracellular membrane of the flagella, midpiece, and, albeit less frequently, on the acrosomal cap of human spermatozoa [10]. Intriguingly, Hartmann and colleagues detected SCFAs in human follicular fluid and vaginal secretion by gas chromatography–olfactometry. In particular, AA was observed in both biological fluids while PA was only in follicular fluid [18].

As a further demonstration of PA presence in the female reproductive tract, *vaginimicrobium propionicum* (VP), a Gram-stain positive anaerobic rod-shaped bacterium, was isolated using culturomics from vaginal discharge [22]. VP belongs to the Propionibacteria family, which are producers of PA by definition. Despite the fact that these results provide evidence that SCFAs reside within gynecological fluids, the data in the literature is still missing about their specific role in human reproduction. Since OR51E2 has been observed on the surface of spermatozoa and it has been shown to bind PA in other cell types, we wondered if OR51E2/PA has a functional role in sperm activation. Additionally, OR51E2 could be involved in the interplay between microbiota and human reproduction as mediators of spermatozoa chemotaxis. The aim of the present study is to gain an insight into the role of OR51E2 as a player of the sperm chemotactic process through its binding to specific ligands in SCFAs.

## 2. Results

### 2.1. Activation of Spermatozoa by SCFAs

Immunofluorescence analysis revealed that OR51E2 was present in the midpiece and partially on the acrosome of non-treated sperm cells, purified by a percoll gradient (Figure 1a). The acrosomal localization was evaluated using the acrosomal peanut agglutinin (PNA) marker. After an exposition of 50 mM of PA for 10 min, we observed an enrichment of OR staining in the acrosomal compartments that attained an average of 40% (Figure 1a,b), indicating that PA was able to mediate OR51E2 acrosomal relocalization. We verified the specificity of the α-OR51E2 antibody overexpressing FlagRho-OR51E2 in HeLa cells and analyzed the protein expression and localization by immunofluorescence (Appendix A).

As a well-known signal of the sperm cell’s activation, pERK levels were measured by Western blot after the PA stimulus in purified spermatozoa. We observed a pERK accumulation in a dose-dependent manner, with a 50 mM PA dose that attained statistical significance (Figure 1c,d). Furthermore, we performed time-course experiments for 5′ and 10′ minutes that revealed a time-dependent accumulation of ERK protein phosphorylation upon PA treatment, too (Figure 1e,f). Moreover, we found that 5 mM AA treatment was able to mediate pERK accumulation, too (Appendix A). Finally, we confirmed pERK increase in sperm cells upon 500 µM of bourgeonal treatment, as previously reported [11] (Appendix A).

To gain further insight into the OR51E2 signaling pathway after PA stimulus, we evaluated Ca^2+^ influx. We added the fluorescent Ca^2+^ indicator to seminal fluid and stimulated the cells with a rapid application of 50 mM PA. To demonstrate the extent of calcium influx, we immediately recorded the fluorescence signal. At both the intermediate and final recording times, we observed that the acute injection of PA caused a slight, but significant, increase in intracellular Ca^2+^ in comparison to the non-treated sperm (Figure 1g). As a positive control, we quantified the calcium signal upon bourgeonal (500 µM) treatment and found that it evoked a similar intracellular Ca^2+^ signal, albeit to a greater extent (Figure 1g). Moreover, to confirm the molecular activation of OR51E2 by PA, we evaluated the cAMP levels upon PA treatment, since it has been reported in the involvement of adenylate cyclase in sperm chemotaxis [2]. As shown in Appendix A, PA stimulus was able to increase the cAMP amount at both 5 and 50 mM doses. Overall, these results show that propionate-mediated sperm molecular activation involves ERK phosphorylation, calcium release, and cAMP increase, which correlates to OR51E2 acrosomal relocalization.

### 2.2. Spermatozoa Activation Mediated by PA Depends on OR51E2 

To address whether propionate-mediated sperm activation depends on the binding to OR51E2, seminal fluid was pretreated with the specific antibody against OR51E2. As a control, seminal fluid was pretreated with an antibody against OR6B2, another olfactory receptor unresponsive to PA [23]. At both the time points analyzed, the pre-treatment of sperm with α-OR51E2 antibody significantly reduced the PA-mediated Ca^2+^ signal in comparison to the pre-treatment with α-OR6B2 (Figure 2a). To confirm the role of ERK phosphorylation in the aforementioned pathway, we analyzed by Western blot the pERK levels upon PA treatment with α-OR51E2 or α-OR6B2 antibodies pre-incubation. As shown in Figure 2b,c, we detected a time-dependent accumulation of pERK, impaired by seminal fluid pre-incubation with the α-OR51E2 antibody (Figure 2b,c).

### 2.3. SCFAs-Induced Sperm Migration

We wondered if PA-dependent sperm activation through OR51E2 acrosomal clustarization could be involved in sperm chemotaxis. To address this question, we performed a modified, direct-swim-up assay. Briefly, we added or not PA and AA to the medium stratified over the seminal fluid, testing their ability to act as chemoattractants for sperm cells in comparison to BSA used as a positive control. We observed a significant increase in migrated spermatozoa in both the PA- and AA-treated samples compared to the control to a greater extent than the BSA treatment (Figure 3a).

We then made use of the CASA system to analyze the kinematic parameters in the swim-up migrated fraction of spermatozoa. We observed statistically significant higher values for some motility parameters, in particular, linearity (LIN) and straightness index (STR), in sperm cells exposed to PA and AA in comparison with the control medium (Figure 3b,c, respectively). On the contrary, we did not observe significant changes in velocity along the average path (VAP) and velocity along the curvilinear path (VCL) (Figure 3b,c, respectively). 

### 2.4. Cervical Mucus Contains SCFAs and Evokes Sperm-Cell Activation

We wondered whether PA was present in cervical mucus, since it was previously reported in the gas chromatography analysis of different fluids of the female reproductive tract, such as intravaginal and follicular fluids [18]. We used liquid chromatography coupled to mass spectrometry to analyze the specific presence of PA in the cervical mucus of fertile women, and we found it in all the samples analyzed (n = 10) (Figure 4a). Moreover, we also detected AA in the same samples (Figure 4b). The presence of these OR51E2 ligands in cervical mucus samples prompted us to investigate the sperm activation after the in vitro incubation of spermatozoa with mucus itself. To this aim, we measured by Western blot the pERK levels in sperm cells incubated with three samples of the cervical mucus analyzed in HPLC for the PA content. As shown in Figure 4c,d, all the three mucus samples are able to trigger pERK accumulation in comparison to untreated sperm cells, in a significant way.

## 3. Discussion

The observation that olfactory receptors are expressed in male and female reproductive systems could uncover how environmental components, represented by active ligands for the identified chemosensory receptors, affect successful reproduction.

The non-orthotopic expression of ORs, meant as the expression outside the olfactory epithelium, has been reported to date in many different organs and in the testis, too [4]. Previous research from our group identified more than 20 ORs in sperm cells, playing roles in the guidance of spermatozoa towards the oocyte, at multiple levels [9]. Furthermore, we showed the presence of different ORs in the ducts of the testis and epididymis [9]. The presence of these receptors on mature cells could be related to sperm motility, while their presence in testicular tubules and epididymis might suggest specific roles in the regulation of sperm maturation and migration. Moreover, DNA sequencing of human testicular biopsies showed the expression of a human olfactory receptor, OR17-4, in both the testis and surface of mature spermatozoa. This OR was reported to influence the chemotactic movement of sperm cells in response to the odorant bourgeonal [14]. A major limit in the proof of concept for the pro-chemotactic role of bourgeonal is that it has not been reported any endogenous corresponding molecule for this odorant [24]. 

Indeed, although new information has emerged concerning the profile of distinct olfactory receptors and coupled molecular pathways in human reproductive systems, endogenous ligands of specific chemoreceptors have only been rudimentarily explored [1]. In this study, we reported for the first time the relocalization of OR51E2 on a human sperm surface and the consequent molecular activation upon SCFAs treatment, in particular, PA. This in turn is related to sperm migration towards SCFA-enriched medium. Moreover, we evidenced the presence of PA and AA in cervical mucus and showed that an incubation of these samples with spermatozoa is able to trigger pERK accumulation, similar to PA and AA treatments. Consequently, due to the exclusive SCFA production by microbiota metabolism, the presence of SCFAs in the female genital tract could suggest a key role of microbiota in fertility and reproduction through, at least in part, the ORs. Similarly, the possible presence of SCFAs in the male reproductive system might also be involved in the regulation of sperm maturation and migration. 

Previously, it was reported that ORs are specifically localized on distinct spermatozoon anatomical areas [10,11]. Flegel and colleagues showed that OR51E2 is expressed on the flagella, midpiece, and, occasionally, acrosomal cap [10]. The authors concluded that different OR localizations can subtend different functions. We found that, upon PA stimulus, OR51E2 was able to relocalize from the flagella and midpiece to the acrosomal cap, just as it was previously reported for OR17-4 in response to the odorant bourgeonal [11]. The OR51E2 acrosomal relocalization upon ligand stimulus suggests this receptor’s involvement in sperm chemotaxis towards the oocyte. 

At a molecular level, it was reported that the process of chemotaxis requires the activation of several pathways. In particular, Teves and colleagues showed a temporal sequence of chemotactic signaling events in which the activation of adenylate cyclase was followed by a cascade of protein tyrosine phosphorylation and calcium influx [24]. 

Within the protein-phosphorylation cascade, ERK phosphorylation (pERK) is a well-established signal for sperm activation [25,26]. We analyzed pERK levels in sperm cells upon PA stimulus and we found a dose and time-dependent accumulation of ERK protein phosphorylation, indicating that the OR51E2 acrosomal clusterization upon ligand binding could subtend a molecular cascade of sperm-cell activation.

We then assessed whether PA stimulus could lead to a Ca^2+^ influx, which has been previously reported to be the key event in the activation of sperm chemotaxis [24,27]. In particular, its role as a downstream molecular effector of ORs–ligand binding should be noted [14,15]. Indeed, Spehr et al. demonstrated that the OR17-4 ligand, bourgeonal, can evoke calcium release and subsequently, sperm chemotaxis [14]. We found that PA stimulus was able to induce a significant increase in intracellular Ca^2+^ in comparison to the control, confirming the involvement of this central molecular signature in the SCFA-mediated activation of sperm cells.

Overall, these results support the hypothesis that PA, through OR51E2 acrosomal relocalization, is able to mediate molecular activation in sperm cells, involving ERK and calcium influx. We demonstrated that spermatozoa activation mediated by PA relies on OR51E2, because the adoption of OR51E2-specific antibody pre-treatment interferes with the previously described pathway. Both pERK accumulation and calcium influx promoted by PA stimulus were significantly reduced with α-OR51E2 antibody pretreatment, supporting the idea that the PA-mediated molecular activation of spermatozoa depends on the binding of PA to this olfactory receptor. 

As stated above, the PA-mediated OR51E2 acrosomal clusterization on the outermost part of the sperm head suggests that this ligand–receptor binding could be involved in the sperm migratory response. To further address the involvement of PA-OR51E2 binding in sperm migration, we performed a modified direct-swim-up assay. Furthermore, we analyzed the differential spermatozoa migration towards a medium with or without PA or AA. In this model, higher quantities of sperm cells in both PA- and AA-enriched mediums were recovered in comparison to the control one. This result suggests a key role for SCFAs in sperm chemotaxis. The CASA analysis of kinematics parameters quantified the phenomenon. Indeed, we found higher VSL and directionality-related parameters, such as LIN and STR, upon SCFA addition. Moreover, we observed that VAP and VCL did not increase in the spermatozoa fraction migrated to PA- and AA-enriched mediums. VSL is the velocity (µm/s) recorded along the straight-line path of the spermatozoon, from the first to the last point of the recorded path itself; on the contrary, VCL is calculated as the velocity (µm/s) through the curvilinear path performed by the sperm cell. LIN is calculated as the percentage ratio VSL/VCL [28]. Therefore, a higher percentage of straight-directed spermatozoa, instead of curvilinear ones, increased LIN. Indeed, a sperm cell characterized by a circling path would have low LIN due to a higher curvilinear path than a straight one [29]. STR, which represents the straightness index, defined by the ratio VSL/VAP, increased in PA-treated spermatozoa as they were characterized by a straighter trajectory along the average recorded path. Therefore, the increase in LIN and STR, related to the increase in VSL, indicated that PA and AA not only promoted a general migration of sperm cells, but specifically a more linear orientation. These data fit with SCFA’s possible role as a chemoattractant through the activation of OR51E2. On the other hand, OR51E2 has been demonstrated to bind SCFA as a specific ligand able to promote downstream molecular activation in many different cellular models [20,30,31]; intriguingly, SCFAs, such as PA and AA, are human endogenous compounds produced by gut microbiota metabolism, through dietary non-digestible carbohydrate fermentation [32,33]. Beyond gut microbiota, its presence has been reported in a vaginal swab from a woman without vaginosis of specific bacterial species producing propionic acid, the vaginomicrobium propionicum [22]. The authors reported propionic and acetic acids as the major end products of bacterium metabolism. This novel microbial identification revealed the local presence of specific bacteria-producing SCFAs as part of the vaginal microbiota. Moreover, Hartmann and colleagues applied gas chromatography–olfactometry coupled to mass spectrometry analysis of both vaginal secretion and follicular fluid aiming to identify odorous compounds able to promote an in vitro activation of three different ORs (OR4D1, OR7A5, and OR1D2) [18]. 

## 4. Materials and Methods

### 4.1. Sperm Samples: Ethical Approval

Sperm collection and analysis were approved by the local ethics committee of the Fondazione Policlinico Universitario A. Gemelli, IRCCS, Rome, Italy (protocol number ID 3943). Human sperm were freshly obtained by masturbation (3–5 days of sexual abstinence) from young, healthy donors who provided informed, signed consent. All samples fulfilled WHO’s 2021 semen-quality criteria for cell count (concentration > 15 million/mL), motility (motile cells > 40%), and viability (live cells > 58%). 

### 4.2. Biological Preparations 

The sperm were handled and prepared as previously described [34,35]. Briefly, liquefied semen was overlaid on a 50% Percoll (cell-culture tested, Sigma-Aldrich, MO, USA) density gradient and centrifuged at room temperature for 30 min at 400 g. The recovered sperm cells were then resuspended in 1 mL of phosphate-buffered saline (PBS; Sigma-Aldrich) and subjected to a residual leukocyte depletion using Dynabeads^TM^ CD45 (Invitrogen, Carlsbad, CA, USA). Then, the samples were washed twice and the pellet was collected, washed in Ringer’s solution (140 mM NaCl, 5 mM KCl, 2 mM CaCl_2_, 2 mM MgCl_2_, 10 mM Hepes, 10 mM glucose, pH 7.4), and again centrifuged for 15 min. Then, the motile spermatozoa pellet was resuspended in Ringer’s solution and used for further experiments.

### 4.3. Analysis of OR51E2 Expression and Localization 

Immunofluorescence and confocal microscopy were used to detect the expression and localization of OR51E2 in human spermatozoa and HeLa cells. In HeLa cells (kindly provided by Dr. Fabiola Moretti), we overexpressed FlagRho-OR51E2 (kindly provided by Dr. Jennifer Pluznick) to verify the specificity of the α-OR51E2 antibody. Human sperm cells, purified as described above, were treated with PA 50 mM or Ringer’s solution (CTR) for 10 min at room temperature. Then, the sperm cells were washed and smeared on polylysine slides, and then allowed to air-dry. Then, HeLa and sperm cells were fixed in 3.7% formaldehyde for 15 min at room temperature (RT) and subsequently treated with 5% BSA to block nonspecific binding. Then, both cellular models were treated with polyclonal anti-OR51E2 (1:200) overnight (Aviva System Biology, San Diego, CA, USA) antibody, washed three times by PBS, and incubated for 45 min with goat Alexa Fluo-488 anti-rabbit IgG (A11034 Molecular Probes, Thermo Fisher, Waltham, MA, USA). For the HeLa cells, DNA was stained with Prolong Gold DAPI (P36935 Molecular Probes), while in sperm cells, acrosomal compartment was counterstained with lectin PNA (L32459 Molecular Probes) following the manufacturer’s instructions. 

### 4.4. Confocal Microscopy

The cells were imaged with an inverted confocal microscope (Nikon A1-MP). Fluorescence images (excitation: 402 nm for the blue channel, 488 nm for green channel, and 561 nm for red channel) were collected in three separated channels (emission filter: 450/50 nm for the blue channel, 525/50 nm for the green channel, and 595/50 nm for the red channel) using a 60× immersion-oil objective with 1.4 Numerical Aperture (NA). Internal photon multiplier tubes collected 2048 × 2048-pixel images in 16 bit at a 0.063 ms dwell time.

### 4.5. Colocalization Analysis

To quantify the compartmentalization of OR51E2 within the acrosome of different spermatozoa, we applied a colocalization approach. Colocalization analysis was performed through the Colocalization Threshold plugin available via the open source software ImageJ (NIH). The analysis of fluorescence colocalization was graphically represented in scatterplots, where the intensity of the green channel was plotted versus the intensity of the red channel for each pixel. A proportional fluorescence intensity of the two probes resulted in the distribution of points along a straight line, with the slope reflecting the ratio of the fluorescence of the two probes. To quantify the intranuclear compartmentalization of OR51E2, we evaluated the Manders’ Colocalization Coefficients. In particular, for two probes, denoted as *B* and *G*, respectively, the coefficient *M_1_* provides the fraction of *B*, i.e., the fraction of pixels expressing the OR51E2 protein, in compartments containing *G*, which is PNA (a marker of the acrosome), according to the following expression:$$M_{1}=\frac{\sum_{i}B_{i,colocal}}{\sum_{i}B_{i}}\;\mathrm{where}\;\{\begin{array}{c} B_{i,colocal}=B_{i}\;if\;G_{i}>0\\ B_{i,colocal}=0\;if\;G_{i}=0 \end{array}$$

The threshold value was automatically identified by applying the Costes method [36] and the coefficient was only evaluated for pixels above threshold (*tM_1_*), with values ranging from 1, when the protein was expressed in the whole acrosome, to 0, in case of no expression of the protein.

### 4.6. Western Blot 

For the Western blot analysis, after PA treatment (5 and 50 mM for 5, 10, and 15 min) the sperm was lysed in RIPA buffer (50 mM Tris–Cl, pH 7.5, 150 mM NaCl, 1% Nonidet P-40, 0.5% Na deoxycholate, 0.1% SDS, 1 mM EDTA) and sonicated for 15 s at 20% intensity and 4 °C, and placed on ice for 30 min. The samples were then centrifuged for 15 min at 15,000× *g* and 4 °C, and supernatant containing proteins were then heat-denatured at 100 °C for 5 min in reducing Laemmli sample buffer 4× (BioRad, Philadelphia, PA, USA). Proteins were resolved by SDS-PAGE and then transferred onto PVDF membranes (Millipore, Burlington, MA, USA). All buffers contained a cocktail of protease inhibitors (Boehringer, Ingelheim am Rhein, Germany). Membranes were developed using enhanced chemiluminescence (ECL westar, Cynagen, Bologna, Italy). Bands were analyzed by a chemiluminescence imaging system, Alliance 2.7 (UVITEC, Cambridge, UK), and quantified by the software Alliance V_1607. The following primary antibodies were used: rabbit α-pERK (cell signaling 9101S) and rabbit α-ERK tot (cell signaling 9102S). In some experiments, to assess the specificity of ERK phosphorylation in response to PA binding to its specific receptor, the sperm cells were pre-treated with polyclonal antibody anti-OR51E2 (OAAF-05032 Aviva System Biology, San Diego, CA, USA) or with non-related antibody polyclonal anti-OR6B2 (NBP2-13711 Novus Biologicals, Milan, Italy) for 10 min at room temperature.

### 4.7. Measurement of Changes in [Ca^2+^]

The changes in the free intracellular Ca^2+^ concentration [Ca^2+^] in human sperm cells were measured in 96 multi-well plates in a fluorescence plate reader (Varioskan Lux, ThermoFisher, Walthman, MA, USA). Briefly, an aliquot of seminal fluids was incubated with the fluorescent Ca^2+^ indicator Fluo-4, AM (10 μM, Fluo-4 Direct Calcium Assay kit, ThermoFisher, Walthman, MA, USA) for 30 min at 37°C. Fluorescence was excited at 480 nm and emission was recorded at 520 nm with bottom optics. Fluorescence was recorded before and after, with or without the addition of compounds (PA 50 mM or bourgeonal 500 µM) and controls to duplicate wells. The changes in Fluo-4 fluorescence were shown as relative fluorescent signals (RFSs). The change in fluorescence signals was evaluated with respect to the basal fluorescence before the addition of compounds. In a set of experiments, to evaluate the specificity of the [Ca^2+^] increase upon PA binding to its specific receptor, the sperm cells were pre-treated with polyclonal antibody anti-OR51E2 or with non-related polyclonal antibody polyclonal anti-OR6B2 for 10 min at room temperature. The changes in Fluo-4 fluorescence were shown as relative fluorescent signals (RFSs). The change in fluorescence signals was evaluated with respect to the basal fluorescence before the addition of compounds.

### 4.8. Adenosine 3′5′-Cyclic Monophosphate (cAMP) Assay

The determination of cAMP levels was performed using the Parameter^tm^ cAMP assay (R&D Systems^®^, Inc., Minneapolis, MN, USA) according to the manufacturer’s instructions. Briefly, purified sperm cells were treated with 5 or 50 mM of PA for 15 min, and were then washed 3 times in cold PBS and resuspended in cell lysis buffer to a concentration of 1 × 10^7^ cells/mL. Cells were subjected to three repeated cycles of freeze/thaw at −20 °C. Lysates were centrifuged at 600× *g* for 10 min at 4 °C to remove cellular debris. The supernatants were immediately assayed for cAMP determination. The absorbance was read at 450 nm. The intensity of the color was inversely proportional to the concentration of cAMP in the sample.

### 4.9. Modified Direct-Swim-Up Assay

After the fluidification of the seminal fluid, the entire volume was divided in fractions of 0.5 mL into round-bottom tubes, and 0.5 mL of culture medium was placed over the semen in each tube, adding or not PA 50 mM, AA 5 mM, or 0.6% BSA, as a known chemoattractant positive control. The tubes were put in the thermo block in a vertical position and incubated at 37 °C for 15 min. After the incubation, 0.2 mL of the upper part of each tube (distal fraction) was gently removed, aspirating the sperm from the upper meniscus downwards [37]. Highly motile sperm cells recovered were counted using a Makler Counting Chamber. 

### 4.10. Evaluation of Sperm-Motility Parameters by CASA

The computer-assisted sperm analysis (CASA) system (Microptic S.L., Barcelona, Spain) was used to analyze sperm motility. Motile spermatozoa obtained from modified swim-up, as described above, were evaluated for each treatment for the following parameters: average path velocity (VAP, µm/s), straight-line velocity (VSL, µm/s), curvilinear velocity (VCL, µm/s), straightness (STR, %), and linearity (LIN, %). Briefly, 3 μL of the sperm sample was pipetted and loaded into a pre-warmed (37 °C) standard-count four-chamber Leja slide (SC 20-01-04-B). To analyze sperm motility parameters, a minimum of 200 sperm at 5 different fields were examined in each group.

### 4.11. UPLC-MS/MS

Two short-chain fatty acids (SCFAs), acetic acid and propionic acid, were measured in cervical mucus samples by ultraperformance liquid-chromatography mass spectrometry (UPLC-MS/MS). After obtaining informed consent, cervical mucus samples were collected from healthy, fertile women attending routine gynecological visits. Cervical mucus was collected by a disposable swab at the ectocervix under vaginal speculum examination before any other operation. All participants voluntarily agreed to participate in this study approved by the local ethics committee of Fondazione Policlinico Universitario A. Gemelli, IRCCS, Rome, Italy—Protocol number ID 3943. These freshly collected cervical mucus samples were kept in sterile tubes, transported to the laboratory on ice, and stored at −80 °C until utilization. The UPLC-MS/MS system consisted of an UPLC and autosampler ExionLC AD system (ABSciex, Framingham, MA, USA) and a Qtrap 6500+ (ABSciex, Framingham, MA, USA) equipped with an electrospray ion source. Analyses were conducted in positive-ion mode. 4-Acetamido-7-mercaptobenzofurazan (AABD-SH) was purchased from Tokyo Chemical Industry (TCI, Nihonbashi-honcho, Tokyo, Japan). Short-chain fatty acids–Mixture 2 (acetic, propionic, isobutyric, n-butyric, 2-methylbutyric, isovaleric, and n-valeric acids) was purchased from Cayman Chemical (Cayman Chemical, East Ellesworth, MI, USA). Dichloromethane, triphenylhphosphine (TPP), 2,2′-Dipyridyl disulfide (DPDS), [^2^H_4_] acetic acid, and [^2^H_3_] propionic acid, were purchased from Sigma (Sigma-Aldrich, St. Louis, MO, USA). Water, acetonitrile, methanol, and formic acid (LC-MS grade) were purchased from Biosolve (Biosolve Chimie, Dieuze, France). The analytical procedure required a deproteinization step, performed by adding 300 µL of acetonitrile containing 15 µg/mL of [^2^H_4_] acetic acid and 15 µg/mL of [^2^H_3_] propionic acid to 100 µL of standard solution or 0.1 g of cervical mucus sample. After vigorous agitation, the sample was centrifuged at 14,000 rpm for 5 min at room temperature; then the supernatant was derivatized. For derivatization, 20 µL of AABD-SH (20 mM in dichloromethane), 20 µL of supernatant, 20 µL of TPP (20 mM in acetonitrile), and 20 µL of DPDS (20 mM in acetonitrile) were added to a glass tube and vigorously vortexed. Derivatization was performed for 15 min at room temperature. The reaction solution was dried under a nitrogen flow and reconstituted with 1 mL of methanol. Samples were loaded onto an ACQUITY PRM BEH C18 1.7 µ 2.1 × 100 mm (Waters Corporation, Milford, MA, USA). The chromatographic separation was performed with a gradient of mobile phases A (water containing 0.1% formic acid) and B (methanol containing 0.1% formic acid), with a flow rate of 0.500 mL/min. The gradient followed this pattern: 0–0.5 min 25% B, 0.5–12.5 min 50% B, 12.51–13.50 min 90% B, and 13.51–15.0 min 25% B. The oven temperature was set at 40 °C. The injection volume was 2 µL, and the total analysis time was 15 min. The optimized parameters for the ion source included: temperature at 450 °C, curtain gas at 20, nebulizing gas (GS1) at 60, drying gas (GS2) at 60, collision-activated-dissociation gas at medium, and ion-spray voltage at 5500 V. The multiple-reaction-monitoring (MRM) transitions for each analyte, their respective collision energy, and cone-voltage values are reported in Table 1.

Data acquisition was performed using mass spectrometer software (Analyst Software 1.7.1, ABSciex, Framingham, MA, USA), while for quantitative analysis we used processing software (Sciex OS, ABSciex, Framingham, MA, USA). 

### 4.12. Statistical Analysis

The results are expressed as values of mean ± standard deviation (SD). The statistical tests used were paired two-tailed Student’s or one-sample *t*-tests when the mean value of the control group was arbitrarily set to one. Significant difference was defined as *p* < 0.05. The software used was GraphPad Prism 7.04.

## 5. Conclusions

This study could shed light on the possible physiological function of chemosensory receptors in successful reproduction. Such knowledge is useful to better understand the pathophysiological role of heterotopic chemosensory receptors in biological systems and could help to develop new strategies for the treatment of infertility. Future functional studies are necessary to both focus on the involvement of individual olfactory receptors in sperm chemotaxis and unravel the roles of SCFAs and microbiota in human fertility.

## Figures and Tables

**Figure 1 ijms-23-12726-f001:**
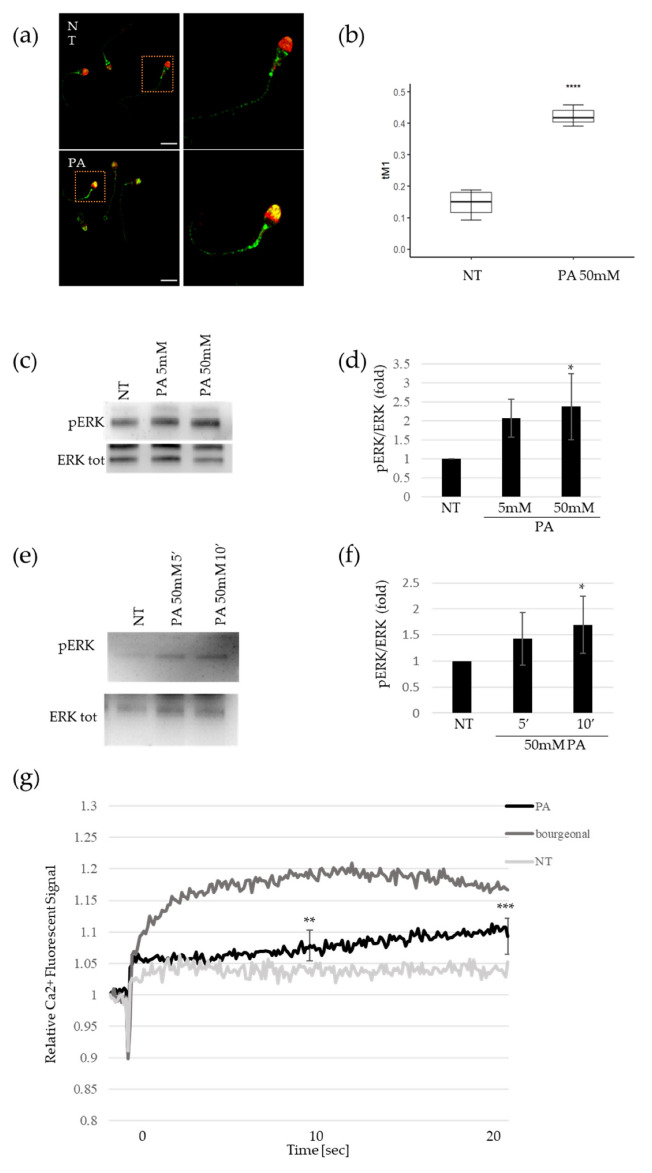
Activation of spermatozoa by propionate. (**a**) Representative pictures of merged confocal immunofluorescence analysis of OR51E2 (green), and acrosomal compartment (counterstained with PNA, shown here in red) in sperm cells before (upper panels, NT) and after 50 mM PA treatment (lower panels, PA). Scale bar: 10 μm. (**b**) Histogram reports the value of tM1 co-localization coefficient of OR51E2-PNA, before and after PA treatment. Ten fields for each slide were counted (n = 30. **** = *p* < 0.0001, two-tailed unpaired *t*-test). (**c**) Representative Western blot (WB) analysis of pERK and total ERK protein in sperm cells treated with 5 and 50 mM PA. (**d**) Histogram shows the ratio of densitometric values of pERK to total ERK in PA dose-dependent experiments. The densitometric value of untreated sperm cells (NT) is arbitrarily set to 1. Mean ± SD of three independent biological replicates are shown (n = 6. * = *p* < 0.05, one sample *t*-test). (**e**) Representative Western blot (WB) analysis of pERK and total ERK protein in sperm cells treated for 5 and 10 min with 50 mM PA. (**f**) Histogram shows the ratio of densitometric values of pERK to total ERK in time–course experiments of PA treatment. The densitometric value of untreated sperm cells (NT) is arbitrarily set to 1. Mean ± SD of three independent biological replicates are shown (n = 3. * = *p* < 0.05, one sample *t*-test). (**g**) Time–course curve of Ca^2+^ influx in seminal fluids treated with 50 mM PA or 500 µM bourgeonal. The fluorescent signal at the time of acute compound injection is set to 1. The fluorescent signals recorded at 10 (middle time point) and 20 s (final time point) are obtained for statistical analysis in comparison to untreated samples (NT). Mean ± SD of six independent experiments are shown (n = 6. ** = *p* < 0.01, *** = *p* < 0.001, one sample *t*-test).

**Figure 2 ijms-23-12726-f002:**
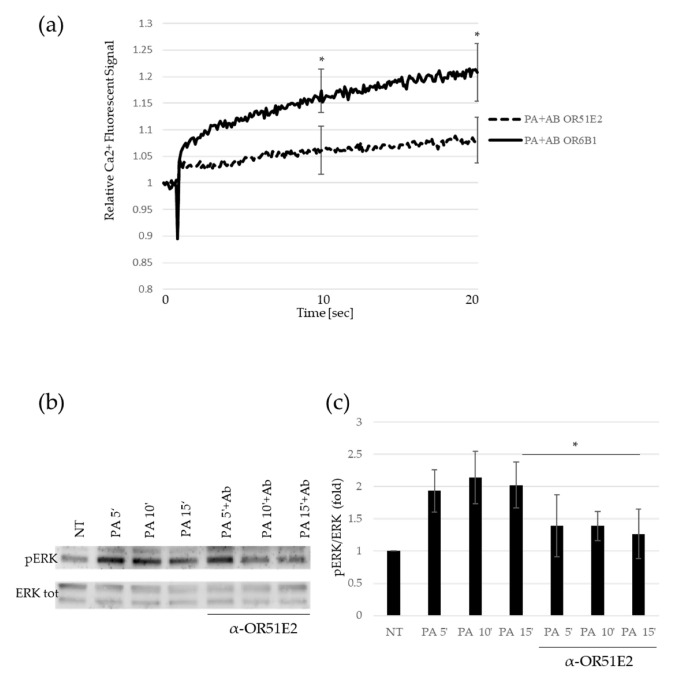
Spermatozoa activation mediated by PA depends on OR51E2. (**a**) Time–course curve of Ca^2+^ influx in seminal fluids preincubated with α-OR51E2 or α-OR6B1 antibodies before 50 mM PA addition. The fluorescent signal at the time of acute PA injection is set to 1. The fluorescent signals recorded at 10 (middle time point) and 20 s (final time point) are obtained for statistical analysis in comparison to untreated samples (NT). Mean ± SD of six independent experiments are shown (n = 6. * = *p* < 0.05, one sample *t*-test). (**b**) Representative Western blot (WB) analysis of pERK and total ERK protein in sperm cells treated in time–course experiments (5, 10, and 15 min) with 50 mM PA pre-incubated or not with α-OR51E2. (**c**) Histogram shows the ratio of densitometric values of pERK to total ERK. The densitometric value of untreated sperm cells (NT) is arbitrarily set to 1. Mean ± SD of three independent biological replicates are shown (n = 3. * = *p* < 0.05, one sample *t*-test).

**Figure 3 ijms-23-12726-f003:**
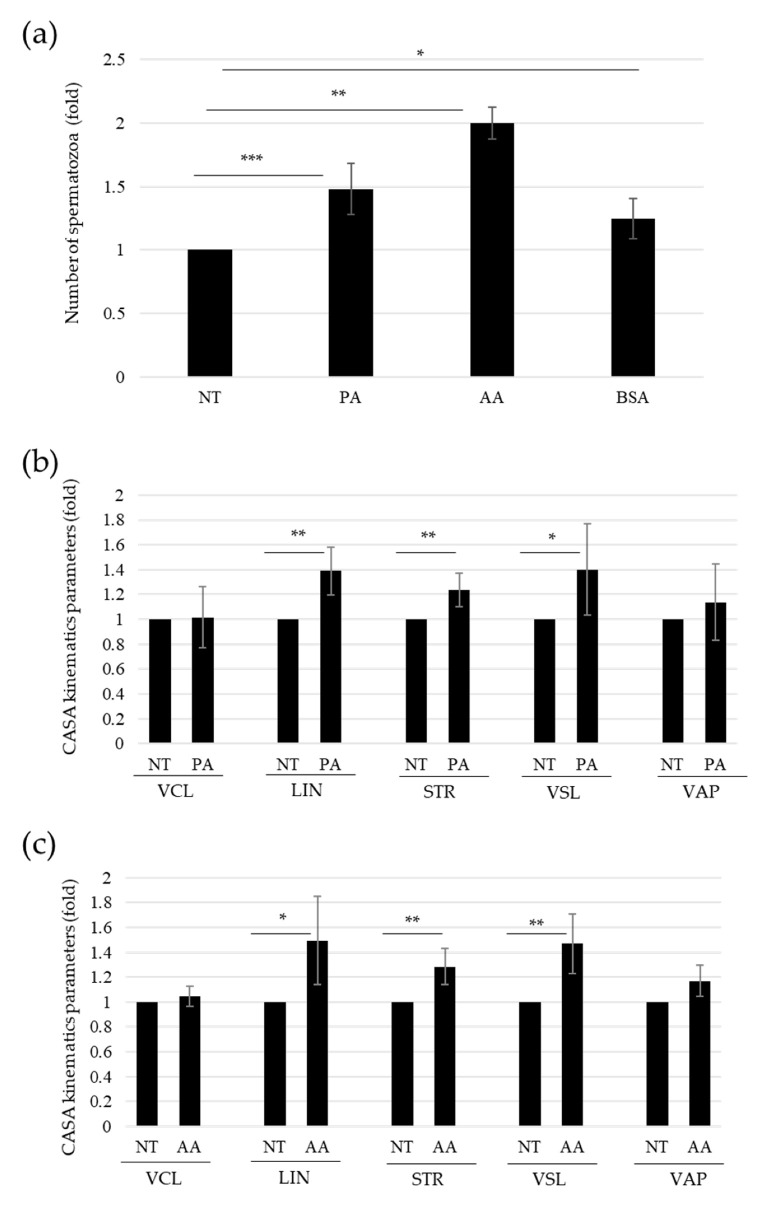
Propionate-induced sperm chemotaxis. (**a**) Histogram shows the numbers of spermatozoa recovered in the distal fractions of modified swim-up assays, upon the indicated treatment (50 mM PA, 5 mM AA, or 0,6% BSA as positive control). The number of sperm cells (NT) recovered in the untreated medium is arbitrarily set to 1. Mean ± SD of six independent experiments are shown (n = 6. * = *p* < 0.05, ** = *p* < 0.01, *** = *p* < 0.001, one sample *t*-test). (**b**) Histogram shows the kinematic parameters obtained by CASA system, relative to sperm cells recovered in the distal fraction of modified swim-up assays, upon the indicated treatment (50 mM PA). Curvilinear velocity (VCL, µm/s), linearity (LIN, %), straightness (STR, %), straight-line velocity (VSL, µm/s), and average path velocity (VAP, µm/s). The kinematic values of sperm cells (NT) recovered in the untreated medium are arbitrarily set to 1. Mean ± SD of 8 independent experiments are shown (n = 8. * = *p* < 0.05, ** = *p* < 0.01, one sample *t*-test). (**c**) Histogram shows the kinematic parameters obtained by CASA system, relative to sperm cells recovered in the distal fraction of modified swim-up assays, upon the indicated treatment (5 mM AA). Curvilinear velocity (VCL, µm/s), linearity (LIN, %), straightness (STR, %), straight-line velocity (VSL, µm/s), and average path velocity (VAP, µm/s). The kinematic values of sperm cells (NT) recovered in the untreated medium are arbitrarily set to 1. Mean ± SD of 8 independent experiments are shown (N = 8. * = *p* < 0.05, ** = *p* < 0.01, one sample *t*-test).

**Figure 4 ijms-23-12726-f004:**
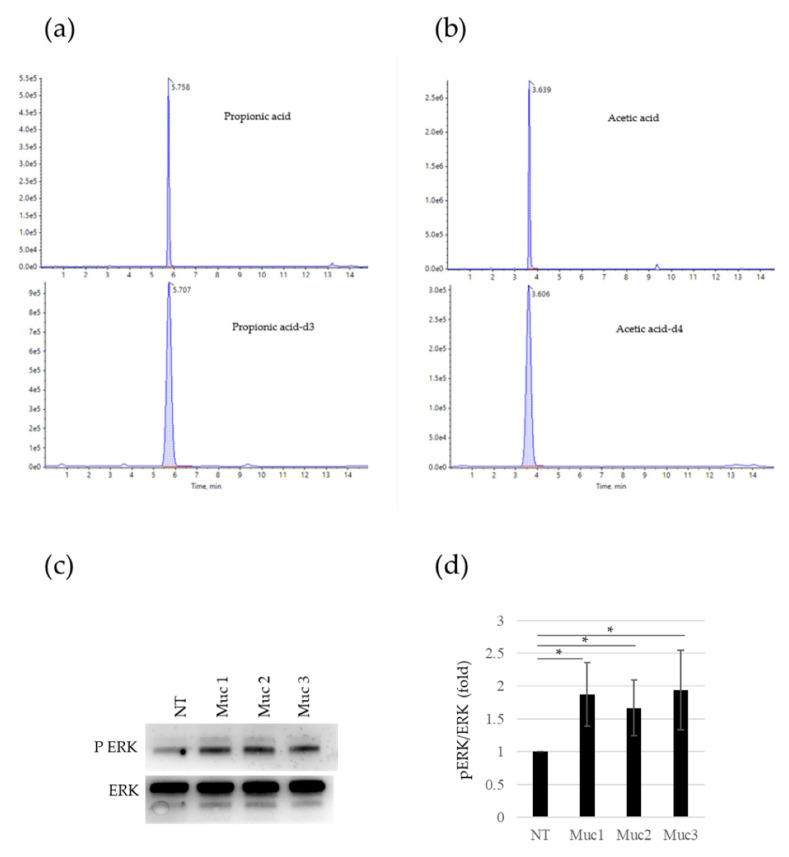
Activation of sperm cells by SCFAs in cervical mucus. (**a**) Plot shows a typical multiple-reaction-monitoring chromatogram of propionic acid (266.1–210.0) at the top and [2H_3_] propionic acid (269.2–210.0) at the bottom of a cervical mucus sample with propionic acid. (**b**) Plot shows a typical multiple-reaction-monitoring chromatogram of acetic acid (255.1–209.9) at the top and [2H_4_] acetic acid (255.1–211.0) at the bottom of a cervical mucus sample with acetic acid. (**c**) Representative Western blot (WB) analysis of pERK and total ERK protein in sperm cells after in vitro incubation for 10 min with cervical mucus obtained from three different healthy, fertile women. (**d**) Histogram shows the ratio of densitometric values of pERK to total ERK in sperm cells after in vitro incubation with cervical mucus. The densitometric value of untreated sperm cells (NT) is arbitrarily set to 1. Mean ± SD of three independent biological replicates are shown (n = 3. * = *p* < 0.05, one sample *t*-test).

**Table 1 ijms-23-12726-t001:** Mass spectrometry conditions.

Compound	Q1 Mass (Da)	Q3 Mass (Da)	DP (Volts)	EP (Volts)	CE (Volts)	CXP (Volts)
Acetic acid	252.1	209.9	40	10	18	10
[^2^H_4_] Acetic acid	255.1	211.0	40	10	22	10
Propionic Acid	266.1	210.0	30	10	18	10
[^2^H_3_] Propionic Acid	269.2	210.0	35	10	22	10

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
