# Peer review of "Short-Chain Fatty Acids Modulate Sperm Migration through Olfactory Receptor 51E2 Activity"

_ijms, 2022, doi:10.3390/ijms232112726_

Round 1

Reviewer 1 Report

The experimental design is good and justify the objectives behind the research. However, in my opinion more olfactory receptors as well as taste receptors could have been included along with the OR51E2 to diversify the research.

Author Response

RESPONSE TO REVIEWER 1

406: Replace conteining with containing

Thanks, we have corrected it.

466-467: Revise the sentence, replace sampling was conducted with samples were collected

Thanks, we have revised it.

468: Replace cervix mouth with external os or Ectocervix

Thanks, we have replaced it.

165: OR6B2 is unresponsive to PA, please add the reference

Thanks for your suggestion; in literature is reported Isobutyraldehyde as specific ligand of OR6B2, a compound not belonging to the SCFAs group. We added the reference (Trimmer C et al., 2018) in the manuscript (lane 165).

Reviewer 2 Report

The study lacks to mention the limitation. Otherwise, the methods are clear and results align with the data. 

In introduction part, the authors need to differentiate proved findings from the hypothesis.  I mentioned with comments, SCFA proved to affect sperm physiology but yet their effect in fertilization to be confirmed.

Either, I propose this study, can further evaluate the effect on fertility, using sperms treated with PA.

Author Response

RESPONSE TO REVIEWER 2

Comment 1, lane 27

Thank you, we mentioned the receptor in full.

Comment 2, lane 47

Thank you we added the reference revising the relative sentence.

Comment 3, lane 58

Thank you for your comment, we modified the sentence.

Comment 4, lane 107

Thank you for the suggestion, we improved the study question.

Comment 5, lane 121

Thank you for the comment; as shown in Fig 1 C, D there is a trend of pERK accumulation upon both PA doses of 5mM and 50mM; in the manuscript we mentioned only 50mM dose because in this condition we reached the statistical significance.

Comment 6, lane 304

Thank you for the comment; overall, the results shown by our research indicate a possible role for OR51E2/SCFAs in the guidance of the spermatozoon in the female genital tract, and therefore it is tempt to speculate a contribution of these players in human reproduction/fertilization.
